# ASyMOB: Algebraic Symbolic Mathematical Operations Benchmark

## Abstract

Large language models (LLMs) are increasingly applied to symbolic mathematics, yet existing evaluations often conflate pattern memorization with genuine reasoning. To address this gap, we present **ASyMOB**, a high-resolution dataset of *35,368* validated symbolic math problems spanning integration, limits, differential equations, series, and hypergeometrics. Unlike prior benchmarks, **ASyMOB** systematically perturbs each seed problem using symbolic, numeric, and equivalence-preserving transformations, enabling a fine-grained assessment of generalization. Our evaluation reveals three key findings: (1) most models' performance collapses under minor perturbations, while frontier systems exhibit substantial robustness, suggesting an emerging *'phase transition'* from memorization to generalization; (2) integrated code tools stabilize performance, particularly for weaker models; and (3) we identify examples where Computer Algebra Systems (CAS) fail while LLMs succeed, as well as problems solved only via a hybrid LLM-CAS approach, highlighting a promising integration frontier. **ASyMOB** serves as a principled diagnostic tool for measuring and accelerating progress toward building verifiable, trustworthy AI for scientific discovery.

## 1 Introduction

In recent years, large language models (LLMs) have shown remarkable capabilities in domains such as mathematical reasoning (Lewkowycz et al. 2022; Kojima et al. 2022; X. Wang et al. 2023; Trinh et al. 2024; Luo et al. 2025; Davies et al. 2021) and code generation (Rozière et al. 2024; Ridnik et al. 2024; Zan et al. 2023; Hou et al. 2024). A crucial skill for real-world applications of these capabilities is mastery of university-level symbolic mathematics, including integration, limit computation, differential equation solving, and algebraic simplification. This proficiency is fundamental across many mathematical, scientific, and engineering challenges.

However, existing mathematical benchmarks inadequately assess symbolic proficiency. Early benchmarks like GSM8K (Cobbe et al. 2021) and MATH (Hendrycks et al. 2021), while driving progress in arithmetic reasoning, focus on pre-university-level questions and have been mastered by frontier LLMs (Glazer et al. 2024). Furthermore, many popular benchmarks rely on multiple-choice questions (Rein et al. 2024), an unrealistic setting which artificially lowers the difficulty. Word-problem benchmarks mix two fundamentally different challenges: text-to-math conversion (understanding the text to build expressions) and symbolic manipulation (solving them). This conflation makes it hard to evaluate an LLM's performance specifically on the latter, and to diagnose the root causes of model errors. Conversely, formal proof datasets (e.g. Zheng et al. 2022; Balunović et al. 2025) address theorem proving but often skip core tasks like integration or solving differential equations.

The broad topic coverage that most benchmarks strive for forces small sample sizes per skill category, hindering robust statistical analysis. For example, in MathBench (H. Liu et al. 2024) only 150 out of 3709 (4%) questions address university-level math in English. The 5K test dataset by Lample et al. (2020) targets symbolic math, but mainly contains simple problems and was immediately saturated. Recent efforts, such as FrontierMath (Glazer et al. 2024) and Humanity's Last Exam (Phan et al. 2025), demand that LLMs exhibit very high proficiency across numerous skills simultaneously, thereby impeding conclusions regarding specific LLM capabilities. Overcoming these

---

Code for ASyMOB dataset generation and LLM evaluation pipeline is attached in the supplementary.

limitations can shed light on a fundamental question: do LLMs solve problems through genuine mathematical understanding or merely through advanced pattern recognition (Mirzadeh et al. 2025; Boye et al. 2025; Z. Zhou, Q. Wang, et al. 2024; K. Huang et al. 2025; Z. Zhou, S. Liu, et al. 2025; Jiang et al. 2024). Addressing this question calls for different types of datasets, which can separate sophisticated pattern memorization from true mathematical abilities.

In response, we present ASyMOB: Algebraic Symbolic Mathematical Operations Benchmark (pronounced Asimov, in tribute to the renowned author). ASyMOB assesses LLM capabilities through systematic perturbations of core symbolic tasks, introducing three key innovations:

1. **Focused Scope**: Targeting pure symbolic manipulation (Figure 1).

2. **Controlled Complexity**: Systematically introduced questions varied by difficulty levels.

3. **High Resolution**: The large scale and fine-grained difficulty steps enable statistically robust measurement of model accuracy, sensitivity to noise types, and impact of tool use.

| **Seed Question** | **Symbolic Perturbation** |
|---|---|
| \<Code / No-Code Prompt\> | \<Code / No-Code Prompt\> |
| *Solve the following integral.* | *Solve the following integral.* |
| | *Assume A, B, F, G are real and positive.* |
| $$\int_1^2 \frac{e^x(x-1)}{x(x+e^x)}dx$$ | $$\int_1^2 \frac{Ae^{Fx}(Fx-1)}{Fx\left(Be^{Fx}+FGx\right)}dx$$ |
| **Solution:** | **Solution:** |
| $$\ln\left(\frac{2+e^2}{2+2e}\right)$$ | $$\frac{A}{BF}\cdot\ln\left(\frac{e^2B+2G}{2(eB+G)}\right)$$ |

| **No-Code Prompt** | *Assume you don't have access to a computer, and do not use code to solve the question.* |
|---|---|
| **Code Prompt** | *Please use Python to solve the question.* |

Figure 1: **Example ASyMOB question and code-use preambles.** A seed question (left) and its symbolically perturbed variant (right). The preamble either disallows or encourages code execution (this part is omitted for models without inherent code execution capabilities).

While there are examples of variational math problem generation (e.g., Mirzadeh et al. 2025; Li et al. 2024), ASyMOB offers three key advancements. First, it evaluates university-level symbolic mathematics - whereas other works remain confined to school-level math, mostly derived from GSM8K and MATH. Second, it introduces entirely new perturbation categories: 'Symbolic' and 'Equivalence' perturbations probe distinct robustness dimensions absent in prior work. Third, it focuses on mathematical reasoning rather than linguistic variation, in contrast to GSM-Symbolic (Mirzadeh et al. 2025), whose perturbations primarily alter textual phrasing and whose most pronounced effects stem from changes in language rather than changes in the underlying mathematics.

Using ASyMOB, we evaluated the performance of open- and closed-weight LLMs, including general and mathematical models. Perturbations significantly challenge LLMs' symbolic math skills, reducing the average model success rate from 74.6% on the unperturbed subset to 46.8% on the full ASyMOB benchmark. Even the simplest perturbations noticeably affect performance (Figure 2).

Following the extensive work on the effects of tool-use in math problem solving (Novikov et al. 2025; Yue et al. 2024; A. Zhou et al. 2024; OpenAI 2025c; Liao et al. 2024; Gou et al. 2024; Imani et al. 2023; Romera-Paredes et al. 2023; Dugan et al. 2024; L. Gao et al. 2023; J. Zhang et al. 2025), we tested code-integrated LLMs both with and without code execution (Figure 2 left) - measuring the effect of tool-use in the specific context of purely symbolic math challenges. Tool-use boosts performance in weaker models, but surprisingly has no positive effect on frontier ones.

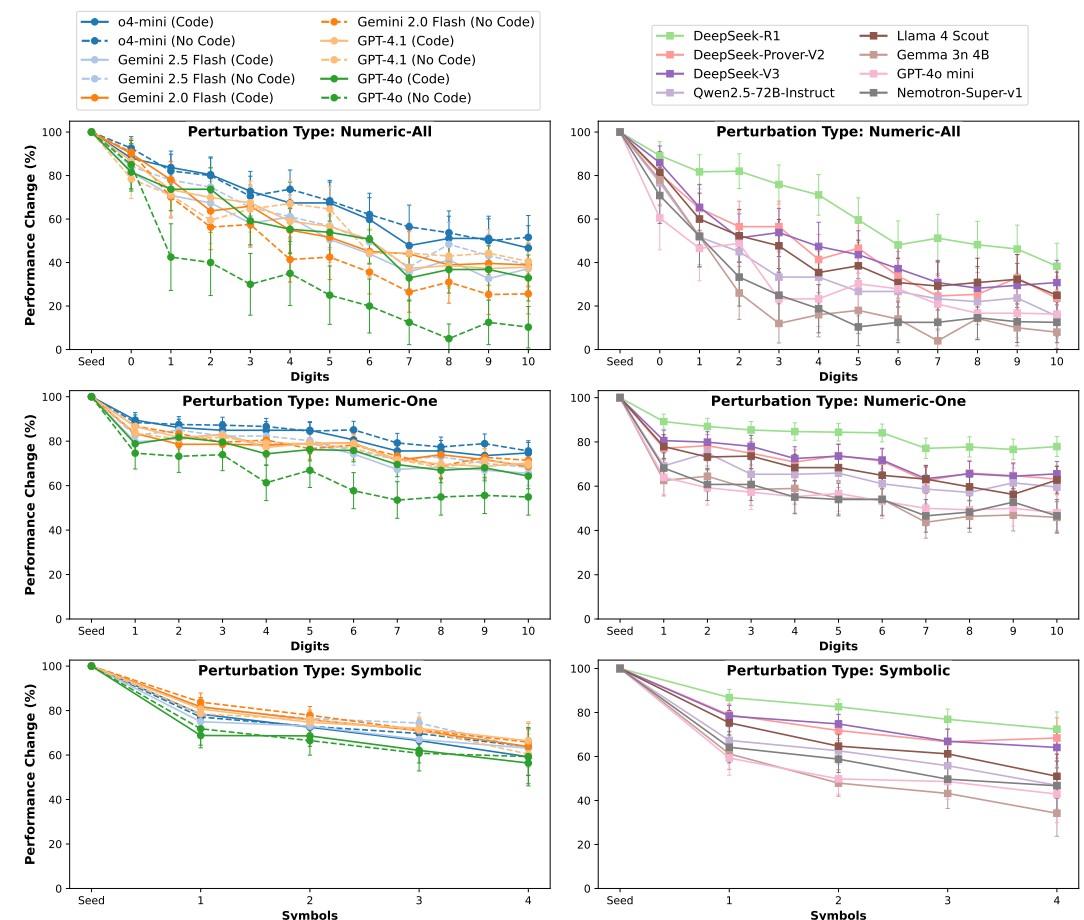

Figure 2: **Degradation of success rate relative to seed-set performance.** Both code-integrated models (left) and non-code integrated (right) exhibit performance degradation due to 'Numeric' and 'Symbolic' perturbations, but frontier models are more resilient. Notably, GPT-4o is substantially more robust when code-enabled. Wald 95% confidence intervals are shown (Wald 1943).

Some perturbed variants in ASyMOB proved impossible for the CAS we tested - Mathematica, WolframAlpha and SymPy (Wolfram Research Inc. 2024; Wolfram Alpha LLC 2025; Meurer et al. 2017) - yet certain LLMs managed to solve them (section 3.1). Moreover, we present an example where pure CAS and pure LLM approaches fail, yet their combination successfully solves the challenge, leveraging the complementary strengths of each system.

## 2 METHODOLOGY FOR SYMBOLIC MATHEMATICAL OPERATIONS MEASUREMENT

### 2.1 DATASET DESIGN AND GENERATION

We begin by curating and creating a set of 100 seed problems that contain only symbolic content - no word-problems or other textual or graphical information beyond the minimal instructions or assumptions needed to define the symbolic task. This restriction excludes almost all olympiad-style problems (B. Gao et al. 2025) and separates our dataset from existing benchmarks. 55 seed questions were curated from university-level benchmarks (Chernyshev et al. 2025; Fang et al. 2024; Frieder et al. 2023; Xu et al. 2025) and math olympiads (Brazilian Mathematical Olympiad 2019; Z. Huang et al. 2024; He et al. 2024). 45 additional seed questions were created to cover underrepresented topics. The questions represent a sample of the practical mathematical challenges that engineers and

scientists frequently encounter in their work and research. Each question is categorized by its topic: Integrals (30), Differential Equations (23), Series (22), Limits (15), Hypergeometrics (10).

Based on these seed questions, we introduce symbolic perturbations to create an overall dataset of 35,368 unique symbolic math challenges (Table 1). The guiding principle was to modify the symbolic structure of the problem - thereby adding a layer of variation - *without substantially altering the core mathematical challenge* or the required solution techniques.

Table 1: **ASyMOB question variants (shown for seed question #6).** For each variant type, the right-most column presents the number of variants for this seed question and the total number of this category in the dataset (e.g. there are 30 'Numeric-One-N' variants of question #6, totaling 3490 'Numeric-One-N' variants over all seed questions). XX, YY, and ZZ in 'Numeric-All-N-S' represent 2 digit random numbers. Full dataset available in the supplementary material.

| Variant | Example Challenge | Answer | # |
|---|---|---|---|
| **Seed (Original)** | $\lim_{x \to 0} \left( \frac{2 \cdot \tan\left( \frac{x}{2} \right)}{x} \right)^{\frac{3}{x^2}}$ | $e^{\frac{1}{4}}$ | 1 (100) |
| **Symbolic-N** (Shown for N=3) | $\lim_{x \to 0} A \cdot \left( \frac{2 \cdot \tan\left( \frac{B \cdot x}{2} \right)}{B \cdot x} \right)^{\frac{C \cdot 3}{(B \cdot x)^2}}$ | $A \cdot e^{\frac{C}{4}}$ | 7 (1348) |
| **Numeric-All-N** (Shown for N=2) | $\lim_{x \to 0} 17 \cdot \left( \frac{2 \cdot \tan\left( \frac{91 \cdot x}{2} \right)}{91 \cdot x} \right)^{\frac{57 \cdot 3}{(91 \cdot x)^2}}$ | $17 \cdot e^{\frac{57}{4}}$ | 11 (1100) |
| **Numeric-One-N** (Shown for N=6) | $\lim_{x \to 0} \left( \frac{2 \cdot \tan\left( \frac{x}{2} \right)}{x} \right)^{\frac{838310 \cdot 3}{x^2}}$ | $e^{\frac{838310}{4}}$ | 30 (3490) |
| **Numeric-All-N-S** (Shown for N=2) | $\lim_{x \to 0} \text{XX} \cdot \left( \frac{2 \cdot \tan\left( \frac{\text{YY} \cdot x}{2} \right)}{\text{YY} \cdot x} \right)^{\frac{\text{ZZ} \cdot 3}{(\text{YY} \cdot x)^2}}$ | $\text{XX} \cdot e^{\frac{\text{ZZ}}{4}}$ | 100 (10000) |
| **Equivalence-One-Easy** | $\lim_{x \to 0^+} \left( \frac{2 \cdot \tan\left( \frac{x}{2} \right)}{x} \right)^{\frac{(\sin^2(-Fx) + \cos^2(Fx)) \cdot 3}{x^2}}$ | $e^{\frac{1}{4}}$ | 15 (1745) |
| **Equivalence-One-Hard** | $\lim_{x \to 0^+} \left( \frac{\sinh\left( \log\left( Ax + \sqrt{A^2 x^2 + 1} \right) \right)}{Ax} \right) \left( \frac{2 \cdot \tan\left( \frac{x}{2} \right)}{x} \right)^{\frac{3}{x^2}}$ | $e^{\frac{1}{4}}$ | 15 (1745) |
| **Equivalence-All-Easy** | $\lim_{x \to 0^+} (\sin^2(-Ax) + \cos^2(Ax)) \left( \frac{2 \cdot \tan\left( \frac{(-\sinh^2(Bx) + \cosh^2(Bx))x}{2} \right)}{(-\sinh^2(Bx) + \cosh^2(Bx))x} \right)^{\frac{(\frac{\ln(x) \cdot \log_x(F)}{\ln(F)}) 3}{((-\sinh^2(Bx) + \cosh^2(Bx))x)^2}}$ | $e^{\frac{1}{4}}$ | 60 (7920) |
| **Equivalence-All-Hard** | $\lim_{x \to 0^+} \left( \frac{\tan(x) + \tan(x(A-1))}{(-\tan(x)\tan(x(A-1))+1)\tan(Ax)} \right) \left( \frac{2 \cdot \tan\left( \frac{\left( \frac{\ln_a\left(\frac{b}{a}\right) + \ln_a(c)}{\log_a(c)} \right)x}{2} \right)}{\left( \frac{\ln_a\left(\frac{b}{a}\right) + \ln_a(c)}{\log_a(c)} \right)x} \right)^{\frac{\left( \frac{r \sum_n \frac{1}{n \ln r}}{1} \right) 3}{\left( \left( \frac{\ln_a\left(\frac{b}{a}\right) + \ln_a(c)}{\log_a(c)} \right)x \right)^2}}$ | $e^{\frac{1}{4}}$ | 60 (7920) |

For instance, consider the elementary integral $\int x^2 e^x dx = e^x \left( x^2 - 2x + 2 \right)$, typically solved using integration by parts.

- An acceptable perturbation is $\int x^2 e^{Fx} dx = \frac{e^{Fx} \left( F^2 x^2 - 2Fx + 2 \right)}{F^3}$. Although this variant introduces a substitution step ($t = Fx$), the fundamental solution technique is preserved.
- Conversely, a modification like $\int x^{2B} e^x dx = (-x)^{-2B} x^{2B} \Gamma(2B + 1, -x)$ would *not* be considered a symbolic *perturbation* as it significantly increases the problem's complexity and demands additional mathematical knowledge compared to the original.

After manually perturbing each seed question with 2-to-5 parameters, additional variants were generated using algorithmic transformations. Note that the random nature of the following question generation methods makes ASyMOB inherently resilient against benchmark hacking and memorization. The dataset can (and should) be re-generated before assessing a new LLM - unlike manual benchmarks which are static and most frontier models were exposed to them during training.

One of the questions we aim to investigate is the effect of the number of symbolic perturbations on model performance. Specifically, we ask whether each additional perturbation further degrades per-

formance, or whether most of the added difficulty for LLMs arises from the introduction of the first symbolic perturbation - transforming the problem to contain non-numeric parameters. To enable this measurement, we systematically remove added symbols from each manually perturbed question, generating all possible combinations. This approach helps avoid subjective bias in perturbation choice. Each variant is labeled as 'Symbolic-N', where $N$ indicates the number of perturbing symbols. For example, a question originally marked as 'Symbolic-4' will yield additional variants: four 'Symbolic-3', six 'Symbolic-2', and four 'Symbolic-1'.

Another key evaluation axis contrasts symbolic and numerical perturbations. Mathematically, if a model can solve a symbolically perturbed question, it should also be able to solve its numeric counterpart via substituting constants by symbols, solved symbolically, and substituted back. Yet, as Figure 2 shows, LLMs often underperform on numeric perturbations compared to symbolic perturbations, suggesting their reasoning remains constrained by their token-based architectures.

To test this, numeric variants were created by replacing every symbolic parameter with a random positive integer of fixed digit length, varying from 0 to 10 digits to probe both in- and out-of-distribution performance (large coefficients being rare in training). Here, 0 digits means replacing all symbols by '1', yielding a mathematically equivalent question - yet, Figure 2 shows even this trivial case degrades performance, further questioning LLMs' true mathematical understanding. These variants are labeled 'Numeric-All-N', where $N$ is the digit length.

Due to the probabilistic nature of LLMs, we measure the stability of mathematical correctness over 50 random variations of 'Numeric-All-N' for $N = 2, 3$ - generating a new set of random 2 or 3-digit numbers per variation (Figure 8 in Appendix C). These variants are marked as 'Numeric-All-N-S'.

To explore whether the initial introduction of a large number causes a disproportionate performance drop, or whether performance declines progressively with each added numeric coefficient, we also create variants where only one symbolic parameter is replaced by a number (ranging from 1 to 10 digits), and the remaining symbols are removed. To avoid selection bias, we generate all possible choices of which symbol to retain and replace. These variants are labeled 'Numeric-One-N'.

Numeric perturbations are similar in spirit to previous works like Mirzadeh et al. (2025), Y. Zhang et al. (2024), Shrestha et al. (2025), Srivastava et al. (2024), and K. Huang et al. (2025) - which are based on GSM8K (Cobbe et al. 2021) or MATH (Hendrycks et al. 2021) word problems, as well as Balunović et al. (2025) - that focuses on constructive proofs. Differing from these previous benchmarks, the larger-scale ASyMOB dataset focuses on advanced symbolic math problems, with no language understanding component, and controlled complexity.

Finally, we evaluate the impact of equivalent-form perturbations. In this case, we complicate the problem by inserting one or more expressions that are mathematically equal to 1. For example, symbol $A$ might be replaced by $\sin^2(-Ax) + \cos^2(Ax)$. While such perturbations (intentionally) introduce extra steps in simplification, the final answer is identical to the original version. We confirmed that the tested LLMs could correctly simplify each expression when presented individually,

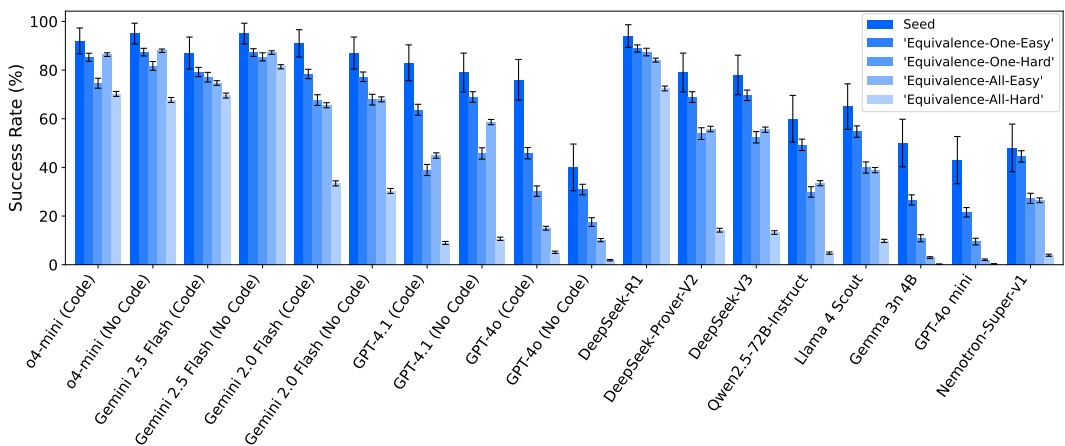

Figure 3: **Effect of 'Equivalence' perturbations.** Note the substantial drop in success rate vs. seed set performance for most models. Wald 95% confidence intervals are shown (Wald 1943).

indicating that the 'Equivalence' variants' additional difficulty arises from the increased structural complexity - not the challenge of simplifying the added expressions themselves.

Five identity types were selected for this transformation - trigonometric, hyperbolic, logarithmic, complex exponential, and series - each with an 'Easy' and a 'Hard' version (see Appendix A.1 for the full list ). The 'Easy'/'Hard' classification was done manually, but the results in Figure 3 retroactively validate our assumptions. To implement this transformation at scale, these identities replace the symbols in the symbolic perturbations. For consistency, each variant uses only 'Easy' or 'Hard' forms. Similar to the numeric case, we generate two types of variants: either all symbols are replaced by equivalent forms ('Equivalence-All-Easy/Hard'), or only one ('Equivalence-One-Easy/Hard').

Some of the resulting expressions may seem "contrived" or unnatural. The 'Hard' perturbations were chosen intentionally to have not just increased difficulty compared to their 'Easy' counterparts, but also be more complex than what is common in questions on conventional exams and in other benchmarks. Part of the goal of such "over-complex" questions is to test LLM ability to *think in steps*. While questions provided in conventional exams would normally not be so complex, intermediate steps in long calculations often create such complex expressions that require symbolic simplification (especially after parameter changes, integration by parts, etc). The ability to simplify complex sub-expressions before attempting to directly solve the complete problem, is itself a skill worth testing.

Notably, SymPy (Meurer et al. 2017) was unable to simplify the more difficult trigonometric and hyperbolic identities to 1, providing an example for CAS limitations in university-level symbolic math challenges.

Figure 3 shows that for most LLMs the challenge level of a single 'Hard' perturbation is lower than multiple 'Easy' perturbations - but not for all LLMs. The reasons behind this difference are a topic for future investigation.

One of the advantages in ASyMOB is once the seed and manual symbolic perturbations are complete and thoroughly validated, all other tasks are generated algorithmically - removing the risk of errors in specific questions or answers. This is not obvious as existing mathematical benchmarks are known to have up to 5-10% mistaken labeling and formatting errors (Vendrow et al. 2024; W. Zhang et al. 2025; Patel et al. 2021). See Appendix A.2 for examples which were discovered during the seed curation process for ASyMOB.

Additionally, by maintaining consistent question formatting and disallowing substantial textual or graphical information, we prevent potential task ambiguities and missing data (Vendrow et al. 2024).

## 2.2 TESTING AND VALIDATION

Validating open-ended symbolic problems is harder than closed-form or numerical ones. For example, the reference answer to question #51 in the ASyMOB dataset is $\frac{1}{2}\sqrt{x}$. However, solving it using Mathematica yields $e^{\frac{1}{2}(\log(x)-2\log(2))}$. Although structurally different, these expressions are mathematically identical. Our evaluation must accept any correct symbolic form and phrasing without penalizing the LLM (e.g. '$\sqrt{x} \cdot \frac{1}{2}$', '$y = \frac{1}{2}\sqrt{x}$', '$y \to \frac{1}{2}\sqrt{x}$', etc.). To prevent false negatives, we implement a multi-step validation process with dual verification methods (see Figure 4).

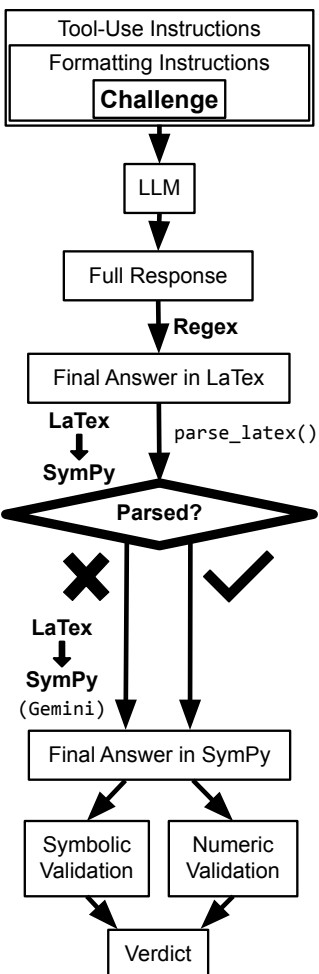

Figure 4: **Result validation pipeline.** The final LaTeX answer is extracted from the full LLM response via a flexible regex. It's parsed into a computable SymPy expression via a deterministic function or, if it fails, via an LLM. The expression is then validated both symbolically and numerically against the reference answer.

The final mathematical answer is extracted from the LLM's full textual response using a highly flexible regular expression (see Appendix B). The extracted LaTeX ex-

pression is then cleaned (e.g. formatting commands like \displaystyle and \boxed are removed) and parsed into a SymPy expression using `sympy.parsing.latex.parse_latex`. If the parsing fails, we resort to using gemini-2.0-flash (Pichai et al. 2024) for this translation (occurred in 18% of cases). Since problem answers are always simpler expressions than the problems themselves, this translation is much easier than the original challenge, and relies on the model's coding skills and not mathematical prowess.

The resulting SymPy expression undergoes two distinct validation checks against the reference answer (also represented as a SymPy object):

- **Symbolic validation**. The difference between the extracted expression and the correct answer is simplified via `SymPy.simplify`. If the simplification reduces this difference to zero (or a constant, in the case of indefinite integrals), the answer is deemed correct.

- **Numeric validation**. We randomly generate numerical values for each variable (e.g., $x$ and any symbolic perturbation parameters) and substitute them into both the LLM's expression and the reference answer. If the relative difference between the two evaluations is less than 0.002%, the answers are considered matching. This process is repeated 5 times to mitigate the risk of coincidental matches. To allow the detection of numeric equivalence between indefinite integrals, we require that all 5 repetitions produce the same difference (not necessarily zero), concluding that the expressions are equivalent up to a constant factor.

This validation approach avoids the need to employ LLMs as judges during evaluation (as was done in Chernyshev et al. 2025 and Fang et al. 2024, among others), thus avoiding validation errors due to LLM pattern recognition biases (as was shown to happen, e.g. in Mao et al. 2024; Chernyshev et al. 2025).

We exclusively use the pass@1 evaluation criterion, reflecting the practical requirement for reliability in real-world applications by engineers and researchers. The inherent LLM randomness is accounted for by evaluating success across the large number of questions within each category.

## 3 EXPERIMENTAL RESULTS

Using the ASyMOB benchmark, open- and closed-weight LLMs were evaluated, including both general-purpose and mathematically-specialized models. Table 2 summarizes their performance.

While frontier closed-weight models (o4-mini, Gemini 2.5 Flash: OpenAI 2025a; Kavukcuoglu 2025) achieve the highest seed accuracy, older (Gemini 2.0 Flash, GPT-4.1, GPT-4o, GPT-4o-mini: Pichai et al. 2024; OpenAI 2025b; OpenAI 2024) and open-weight models (DeepSeek-V3, DeepSeek-R1, DeepSeek-Prover-V2-671B, Llama-4-Scout-17B-16E-Instruct, Qwen2.5-72B-Instruct, Gemma-3n-e4b-it, Llama-3_3-Nemotron-Super-49B-v1: DeepSeek-AI 2025b; DeepSeek-AI 2025a; Ren et al. 2025; Meta 2025; Yang et al. 2024; Farabet et al. 2025; Bercovich et al. 2025) also perform reasonably well, all scoring at least 40%.

A significant finding is the substantial degradation in performance when models are faced with perturbed versions of the seed questions (Figures 2, 3). Some LLMs struggle more with symbolic perturbations, suggesting gaps in mathematical understanding, while others falter with numeric perturbations, possibly due to longer token chains. Understanding the reasons behind these differences between models may reveal deeper principles of how LLMs process mathematical structures.

Where the top models truly shine is their robustness to perturbations - which is arguably a more critical metric for assessing LLM generalization capabilities - netting a performance gap of 20% between o4-mini, Gemini-2.5 Flash, and DeepSeek-R1, to the next best model on the total dataset. This robustness persists across perturbation categories and mathematical topics (Figure 5), even when faced with out-of-distribution challenges, which might indicate a recent 'phase transition' of frontier LLMs from reliance on memorized patterns to genuine mathematical understanding.

Comparing LLM performance on ASyMOB to recent competitive math benchmarks, like AIME2025 (Balunovic et al. 2025) and RIMO (Chen et al. 2025), we see a general correlation. However, we notice that DeepSeek-Prover-V2-671B - despite achieving 88.9% pass ratio on the MiniF2F proof benchmark (Zheng et al. 2022; Ren et al. 2025) and outperforming both Gemini-2.5 and DeepSeek-V3 on PutnamBench (Tsoukalas et al. 2024) - is still surpassed by DeepSeek-R1

Table 2: **Model performance on ASyMOB by perturbation category**. Bold indicates the top performer in each category. Subset titles are color-coded in accordance to Table 1. The bottom line shows SymPy success statistics, providing pure CAS performance baseline. Note that SymPy's 100% success rate on the Seed set is unsurprising, as we validated all questions during seed selection and creation via CAS, introducing a natural selection bias in favor of SymPy.

| Model | Seed | Symbolic | Numeric | Equivalence | Variance | Total. |
|---|---|---|---|---|---|---|
| **Closed-Weights Models** | | | | | | |
| o4-mini (code) | 92 | 69.0 | 74.9 | 78.6 | 72.8 | 76.1 |
| o4-mini (no code) | **95** | 71.8 | **78.1** | 79.0 | **76.8** | 78.1 |
| GPT-4.1 (code) | 83 | 66.1 | 66.3 | 31.3 | 62.8 | 46.2 |
| GPT-4.1 (no code) | 79 | 64.7 | 64.8 | 38.7 | 58.8 | 48.9 |
| GPT-4o (code) | 76 | 57.1 | 61.3 | 15.1 | 59.3 | 35.3 |
| GPT-4o (no code) | 40 | 34.5 | 32.3 | 9.3 | 21.6 | 16.8 |
| GPT-4o-mini | 43 | 26.9 | 27.6 | 3.8 | 17.6 | 11.8 |
| Gemini-2.5 Flash (code) | 87 | 70.3 | 68.2 | 73.2 | 62.6 | 69.5 |
| Gemini-2.5 Flash (no code) | **95** | 75.9 | 72.6 | **84.7** | 69.5 | **78.5** |
| Gemini-2.0 Flash (code) | 91 | 71.9 | 68.2 | 53.7 | 59.7 | 58.1 |
| Gemini-2.0 Flash (no code) | 87 | 69.7 | 64.1 | 53.4 | 51.2 | 54.9 |
| **Open-Weights Models** | | | | | | |
| DeepSeek-V3 | 78 | 64.2 | 59.5 | 39.2 | 48.2 | 45.4 |
| DeepSeek-R1 | 94 | **78.8** | 76.7 | 80.1 | 75.2 | 78.3 |
| DeepSeek-Prover-V2-671B | 79 | 65.6 | 59.8 | 39.8 | 50.1 | 46.3 |
| Llama-4-Scout-17B-16E-Instruct | 65 | 50.6 | 48.2 | 28.5 | 36.7 | 34.3 |
| Qwen2.5-72B-Instruct | 60 | 45.3 | 43.5 | 22.8 | 29.1 | 28.2 |
| Gemma-3n-e4b-it | 50 | 30.4 | 30.3 | 4.7 | 15.1 | 12.0 |
| Nemotron-Super-49B-v1 | 48 | 37.1 | 34.0 | 18.9 | 23.6 | 23.0 |
| SymPy | 100 | 56.7 | 65.2 | 21.9 | 57.8 | 39.2 |

(from the same model family), on every category in ASyMOB. Furthermore, its performance gains vs. the base model (DeepSeek-V3) are incremental at best. This suggests that proficiency in formal proof generation may not directly translate to skill in the broader set of symbolic mathematical operations, where the reasoning capabilities of general models can prove more effective. Nemotron-Super (Bercovich et al. 2025), on the other hand, shows relatively high perturbation resilience, despite the low success rate on the seed subset.

The 'Variance' subset provides insights into model consistency. The variance of results over all 'Numeric-All-N-S' variants was calculated per seed question and per model (Figure 8). An interesting observation is the absence of correlations of variance between models per seed question, indicating that the effect of perturbation is similar regardless of the specific seed (see Appendix C).

Enabling code execution improved the performance of older models (GPT-4o by up to 37.7% and Gemini-2.0 Flash by up to 8.6% in a single category), likely compensating for their symbolic-math

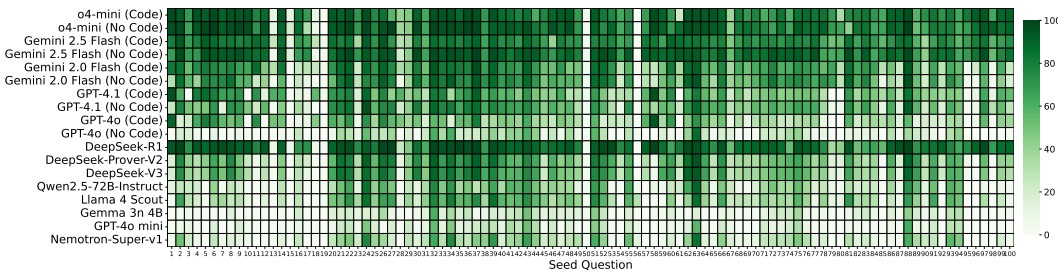

Figure 5: **Heatmap of overall performance per model per seed, averaged over all perturbations.**

weaknesses through coding skills. In contrast, frontier models performed similarly or worse with code execution, likely because their limitations become apparent on the hardest problems - which are usually unsolvable by a naive application of SymPy - so gains require combining the model's internal reasoning (to break down complex problems) with strategic tool use. Both effects highlight the value of hybrid solution strategies.

### 3.1 COMPUTER ALGEBRA SYSTEMS LIMITATIONS

While CAS like SymPy, Mathematica, and WolframAlpha are powerful tools for symbolic mathematics, they have their own limitations. The ASyMOB benchmark includes instances where traditional CAS fail yet LLMs manage. Symbolic perturbations, while apparently easier for LLMs to handle than numeric perturbations, seem to have a much larger detrimental effect on CAS, with multiple examples of CAS solving the seed variant and then failing on a 'Symbolic' variant.

For example, 2 of the 5 'Hard Equivalence' forms (Appendix A.1) are not recognized by SymPy as identical to 1. Yet, many 'Equivalence' variants containing these identities are successfully solved by models in our testing. Another example is the aforementioned ASyMOB question #6 (Table 1) - where WolframAlpha does not simply fail to answer on variant 'Symbolic-3', but produces a false result[1]. Such examples provide added motivation for developing LLMs skillful at symbolic mathematical manipulations, capable of overcoming CAS shortcomings.

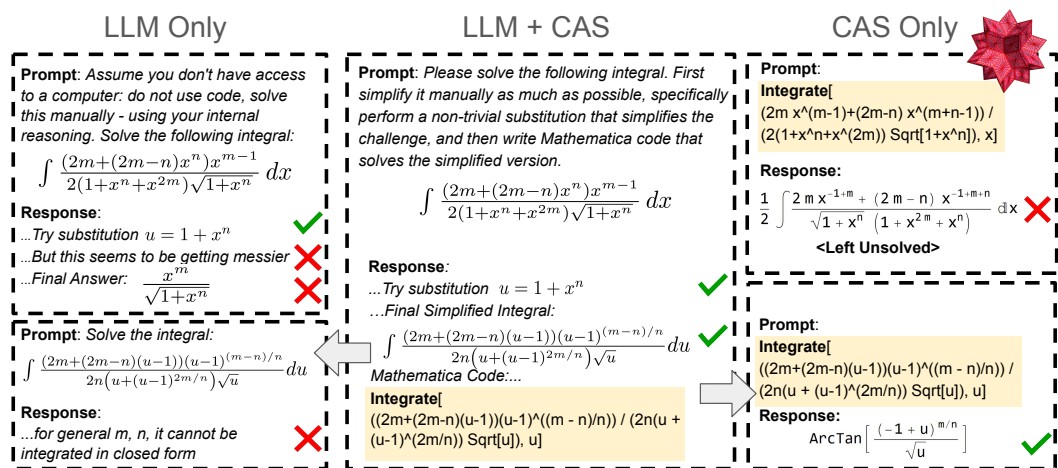

Figure 6: **Example question solved exclusively by a hybrid LLM+CAS approach.** ASyMOB's question #122 was solved incorrectly (left) by GPT-4o, despite the model "considering" a correct substitution. Standard CAS systems also failed to solve the question (right). However, a hybrid strategy succeeded: GPT-4o was prompt to first simplify the problem via substitution and then use CAS code on the simplified expression - enabling Mathematica to solve the question.

Perhaps the most teaching example is ASyMOB question #122 on GPT-4o (Figure 6). Pure CAS and pure LLM approaches both failed. However, when instructed to simplify the integral first and then solve using CAS, the model succeeded, demonstrating the power of combining LLM strategic ability with CAS rigor.

## 4 DISCUSSION AND OUTLOOK

We introduced ASyMOB, a high-resolution symbolic mathematics benchmark that isolates core symbolic reasoning skills, containing 35,368 challenges. Assessment of leading models shows:

---

[1]Tested on Wolfram Language version 14.2.1: `https://www.wolframalpha.com/input?i=Limit%5BA+%28Tan%5B%2B+x%29%2F2%5D%2F%28%28B+x%29%2F2%29%29%5E%28%28C+3%29%2F%28B+x%29%5E2%29%2C+x+-%3E+0%2C+Direction+-%3E+%22FromAbove%22%5D`

- LLMs' symbolic math performance substantially degrades under perturbations, suggesting reliance on pattern memorization and lack of "true understanding".
- Frontier LLMs show a leap in robustness against perturbations of various kinds, suggesting strong symbolic math generalization capabilities.
- Correct tool-use (code execution) can meaningfully improve performance, especially when applied via hybrid LLM+CAS strategies.

Benchmarks aspire to present uncontaminated "new" questions, but ASyMOB bypasses this challenge via systematic perturbations. Even if seed questions are contaminated, the benchmark results remain meaningful - an increasingly important property as sourcing truly novel questions becomes infeasible for large-scale datasets.

To empirically assess this robustness, we ran experiments on Gemini 2.0 Flash, OpenAI GPT-4o, and LLaMA 3.3 Nemotron Super, explicitly including the original seed question and its correct answer as an in-context exemplar within the prompt. While performance improved on simple perturbations (Numeric-All-0: +2%, +27%, +43.5% respectively), the effect quickly dropped on more complex ones (Numeric-One-3, Numeric-All-3, Symbolic-3: +2%, +5.1%, +6.8% respectively). These findings show that seed question contamination does not substantially distort performance on harder variants, and ASyMOB's complex perturbations still expose limitations beyond memorization. Given the extremity of this setup, these modest gains likely represent an upper bound from pretraining, underscoring ASyMOB's robustness in detecting genuine generalization.

Contamination can even be reframed as a feature: if a model leverages prior knowledge of a seed question to solve perturbed variants, it demonstrates real generalization. Eventually, if LLMs improve on re-generated questions by training on earlier iterations, that signifies deeper mathematical understanding - a desirable capability, not a flaw.

Looking forward, LLMs should be intentionally trained to generalize, both via tool use and through systematic perturbations on the training set. Previous works showed that such synthetic data improves overall performance (e.g. Li et al. 2024). Fine-grained perturbations provide a systematic method for generating high-quality synthetic data, offering a valuable resource for fine-tuning future reasoning models.

One of our perturbations is inspired by GSM-Symbolic (Mirzadeh et al. 2025) - which showed that even "trivial" complications in textual math questions can substantially reduce success rates (up to 65%). Similarly, in our work, symbolic complications also led to substantial performance drops (up to 60.9%). This test generalizes the finding of GSM-Symbolic that "current LLMs are not capable of genuine logical reasoning", now shown in the domain of symbolic manipulations and not just in text-to-math conversion.

Importantly, our results suggest a possible solution: once an LLM learns *when* and *where* to use tools, it can mitigate substantial pitfalls by using code execution as a form of grounding. This can be encouraged through prompting strategies like "simplify-then-code" (Figure 6).

Until recently, the hybrid LLM+CAS approach appeared to be the most promising path forward. However, the surprising finding that frontier models *no longer benefit* from CAS use for symbolic math triggers deeper and more fascinating possibilities. Looking ahead, we see three possible trajectories for future developments in AI for math and AI for science:

1. **Intrinsic mastery:** Frontier models may continue to improve in their inherent abilities, eventually surpassing the need for external symbolic math tools, as in the frontier model behavior observed in this work.

2. **Deeper integration:** Tool use may remain essential, but will demand increasingly sophisticated CAS capabilities that co-evolve with LLMs, complementing their inherent abilities and motivating the next generation of CAS infrastructure.

3. **Autonomous tool creation:** LLMs may internalize symbolic computation itself - leveraging their reasoning and coding capacities to build internal, CAS-like mechanisms that blur the boundary between model and tool.

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

## A    ADDITIONAL DETAILS ABOUT THE DATASET

### A.1    LIST OF EQUIVALENCE PERTURBATIONS

The complete list of 'Equivalence' perturbations, discussed in section 2.1, is provided in Table 3.

| Category | Equivalence Perturbation |
|:---:|:---:|
| **Easy** | |
| **Trigonometric** | $\sin^2(-Qx) + \cos^2(Qx)$ |
| **Hyperbolic** | $\cosh^2(Qx) - \sinh^2(Qx)$ |
| **Logarithmic** | $\dfrac{\ln(x)\,\log_x(Q)}{\ln(Q)}$ |
| **Series** | $\dfrac{Q\sum_{N=1}^{\infty}\frac{2^{-N}x}{Q}}{x}$ |
| **Complex Exponential** | $-\dfrac{i(e^{iQx} - e^{-iQx})}{2\sin(Qx)}$ |
| **Hard** | |
| **Trigonometric** | $\dfrac{\tan(x) + \tan(x(Q-1))}{(1 - \tan(x)\tan(x(Q-1)))\tan(Qx)}$ |
| **Hyperbolic** | $\dfrac{\sinh\left(\log(Qx + \sqrt{Q^2x^2 + 1})\right)}{Qx}$ |
| **Logarithmic** | $\dfrac{\log_Q(x/e) + \log_Q(e)}{\log_Q(x)}$ |
| **Series** | $\dfrac{Q\sum_{N=1}^{\infty}\frac{6x}{\pi^2 N^2 Q}}{x}$ |
| **Complex Exponential** | $-\dfrac{2i(e^{4iQx} + 1)\tan(Qx)}{(1 - e^{4iQx})(1 - \tan^2(Qx))}$ |

Table 3: List of the 'Easy' and 'Hard' expressions, which are identical to 1 for any $Q$ and $x$, used in the 'Equivalence' perturbations.

### A.2    DISCOVERED BENCHMARK ERRORS

As mentioned in Section 2.1, existing mathematical benchmarks are known to have up to 5-10% mistaken labeling and formatting errors (Vendrow et al. 2024; W. Zhang et al. 2025; Patel et al. 2021).

For example, question 97 from the GHOSTS 'Symbolic Integration' subset (Frieder et al. 2023): "What is the integral of $2x - x^7\mathrm{atan}(3)$". The output: "...The antiderivative... $\frac{2x^2}{2} - \frac{1}{7}x^8\mathrm{atan}(3) + C$" receives a 5/5 rating, but the $\frac{1}{7}$ should have been $\frac{1}{8}$, potentially creating false positives.

Another example from OlympiadBench (He et al. 2024, subset 'OE_TO_maths_en_COMP', id:2498): "If $\log_2 x - 2\log_2 y = 2$, determine $y$, as a function of $x$". The dataset provides both a full solution: "...to obtain $y = \frac{1}{2}\sqrt{x}$", and a final answer: "$\frac{1}{2}, \sqrt{x}$". The extra comma that appeared in the middle of the final answer prevents deterministic systems from recognizing correct answers.

We inserted both of these questions (with corrected answers) as two of our seeds.

ASyMOB's algorithmic generation methods substantially reduces the risk for such errors in specific questions or answers.

### A.3 'SYMBOLIC-N' SUBSETS ANALYSIS

Due to the requirement that substituting all symbols with 1 reverts the question to its original seed form, the total number of 'Symbolic-N' variations depends on N. For instance, ASyMOB contains only 7 'Symbolic-5' questions. This small sample size is the reason 'Symbolic-5' is not represented in Figure 2, as it is insufficient for robust statistical analysis. This variability also means that the baseline difficulty of 'Symbolic-N' questions changes with different values of N. The 7 seed questions with a maximal perturbation of 5 symbols have an average success rate across all models of 86.6%. In contrast, the 13 seed questions with a maximal perturbation of 4 symbols have a 74.7% success rate, and the overall success rate across all seeds is 73.9%. The 'Symbolic-4' subset includes 13 questions with maximal 'Symbolic' perturbation (derived from the 13 seeds mentioned above) and 35 permutations based on the 7 maximally perturbed 'Symbolic-5' questions. It is likely that the lower initial difficulty of the seeds influences the difficulty of their derived variations to some extent. Therefore, the difficulty of each 'Symbolic' subset should not be assumed to be identical. This effect can account for the slight increase in success rate observed across most models in the bottom graphs of Figure 2 for 3 and 4 symbols.

## B TESTING DETAILS

As noted in section 2.2, a core principle of the test process is to rely on deterministic and predictable tools whenever possible. Figure 4 shows a "Formatting Instructions" wrap around the challenge text. Specifically, these instructions state:

*"Finish your answer by writing "The final answer is:" and then the answer in LaTeX in a new line. Write the answer as a single expression. Do not split your answer to different terms. Use $$ to wrap the LaTeX text. Do not write anything after the LaTeX answer."*

The primary goal is to encourage the LLM to produce a clear LaTeX expression, labeled with "The final answer is:". We opt against using forced structured outputs, even when available, to ensure a fair comparison with models lacking this capability and to avoid introducing requirements beyond symbolic math skills. In essence, we aim to minimize the impact of specific phrasing and structural choices in both language and mathematical presentation.

Once the full answer is received, a series of regexes are used to extract the final answer:

```
Pattern 1 (as instructed):
    r'\**[Tt]he final answer is:?\**\s*'
    r'(?:(?:\\\()|(?:\\\[)|(?:\$+))'
    r'(.*?)'
    r'(?:(?:\\\))|(?:\\\])|(?:\$+))'

Pattern 2 (last boxed expression):
    r'\\boxed\{(.*?)\}' + '(?:\n|$|")'

Pattern 3 (last display expression):
    r"\$+(.*?)\$+"

Pattern 4 (output='  ' case):
    r"output='(.*?)'"
```

```
Pattern 5 (output="   " case):
    r'output="(.*?)"'
```

While the first pattern represents the given formatting instructions - other output formats were accepted as well. It's important to note that responses claiming, for example, the challenge is impossible or asking for specific values to substitute into the symbols, will frequently lack fitting LaTeX expressions. Therefore, the absence of relevant LaTeX usually indicates a missing or incoherent answer, not a parsing issue. Overall, this stage was successful in 98% of cases.

The extracted LaTeX expression is then cleaned and parsed into a SymPy expression using `sympy.parsing.latex.parse_latex`. If the parsing fails, we resort to using an LLM (gemini-2.0-flash) for this translation. It's important to note that not all "final answer" expressions extracted by our permissive regexes are valid LaTeX or even mathematical expressions. Therefore, a failure to produce a working SymPy expression usually indicates a broken or irrelevant answer, rather than a translation issue. Overall, this stage was successful in 96.1% of cases.

As detailed in Section 2.2, the resulting SymPy expression undergoes two distinct validation checks against the reference answer (also represented as a SymPy object). Due to the limitations of SymPy (imperfections in `SymPy.simplify`, handling of very large numbers in `.evalf()`, etc.), if either validation method confirms an answer, it is treated as correct (false positives are highly unlikely). Out of all the valid SymPy expressions created on the previous stage, 97.6% were successfully tested. Responses that could not be verified by either method due to SymPy's technical limitations were excluded from the data analysis and omitted from the reported statistics.

In terms of resources required for this work, by far the largest cost was querying the LLMs. The specific costs per setup (model with and without code execution) are summarized in Table 4. Dataset generation compute was negligible (less than 5 minutes on a single workstation), while the validation stage was more resource-intensive (~10 hours on 3 workstations). Note that the validation process is trivially parallelizable.

| Model Name | Cost |
|---|---|
| **Closed-Weights Models** | |
| Gemini-2.0-flash (no code) | 33$ |
| Gemini-2.0-flash (code) | 32$ |
| Gemini-2.5-flash (no code) | 687$ |
| Gemini-2.5-flash (code) | 648$ |
| GPT-4.1 (no code) | 524$ |
| GPT-4.1 (code) | 1574$ |
| GPT-4o (no code) | 545$ |
| GPT-4o (code) | 1595$ |
| GPT-4o-mini | 16$ |
| o4-mini (no code) | 799$ |
| o4-mini (code) | 1849$ |
| **Open-Weights Models** | |
| Gemma-3n-e4b-it | 22$ |
| Llama-4-Scout-17B-16E-Instruct | 273$ |
| Nemotron-Super-49B-v1 | 250$ |
| Qwen2.5-72B-Instruct | 261$ |
| DeepSeek-Prover-V2-671B | 244$ |
| DeepSeek-R1 | 927$ |
| DeepSeek-V3 | 244$ |

Table 4: Cost of evaluating each test setup on the full ASyMOB dataset. Prices vary mostly due to the vendor and the cost of tool use.

In terms of LLM queries, ASyMOB's testing cost is kept in check by using a single LLM call per problem, unlike many prior works, which assess models using protocols that involve multiple calls per problem. For instance, Lewkowycz et al. 2022 evaluate Minerva on the MATH dataset

(Hendrycks et al. 2021; 12.5K problems) using maj1@k majority voting with k values up to 256, resulting in a minimum of 3.2 million LLM calls. The ASyMOB assessment approach is roughly 90X more cost-efficient due to our use of pass@1. The inherent LLM randomness is accounted for by evaluating success across the large number of questions within each category.

## C  DATA ANALYSIS

Figure 7 presents each model's (with and without code execution) success on each seed question - showing a mix of easier and harder challenges.

Figure 8 illustrates the variance within each 50-question subset of variant 'Numeric-All-2-S' (per seed). Each cell is marked with a 'V' if the model correctly solved at least half of the 'Numeric-All-2-S' variants from that seed question, and an 'X' otherwise.

It is important to note that while correct answers are unique (aside from presentation differences), incorrect answers can vary significantly, including instances where no answer is provided. Consequently, low consistency might result in lower variance for questions with a low success rate compared to those with a high success rate. Indeed, the average variance for all 'V' questions is 0.11, whereas for 'X' questions, it is 0.07.

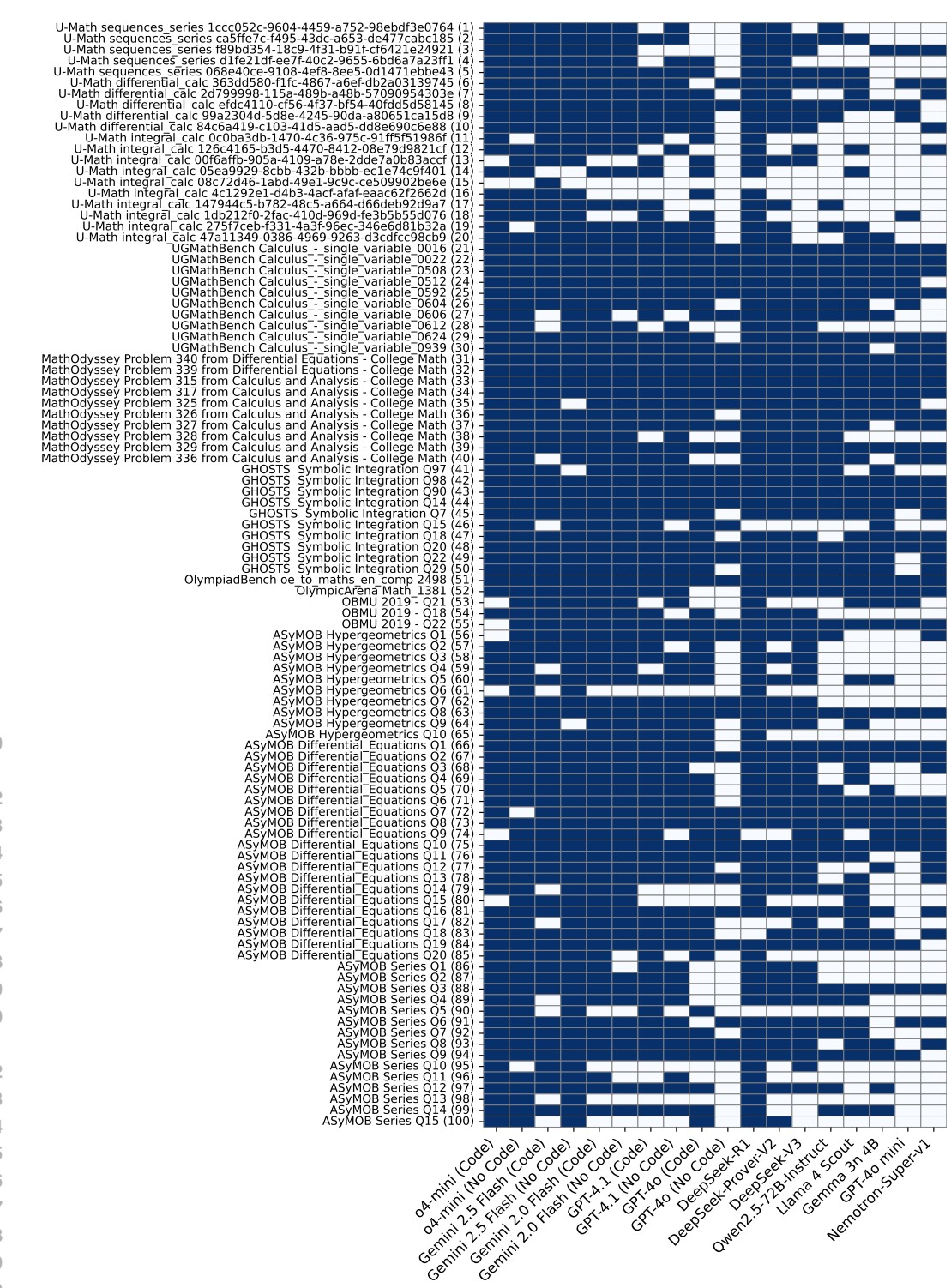

Figure 7: **Model success (blue) / failure (white) per seed question.** Seeds are marked by their source and index in the dataset. Note the difference in challenge level between seeds with different sources. 'ASyMOB' source indicates original questions that were created for the purpose of this work.

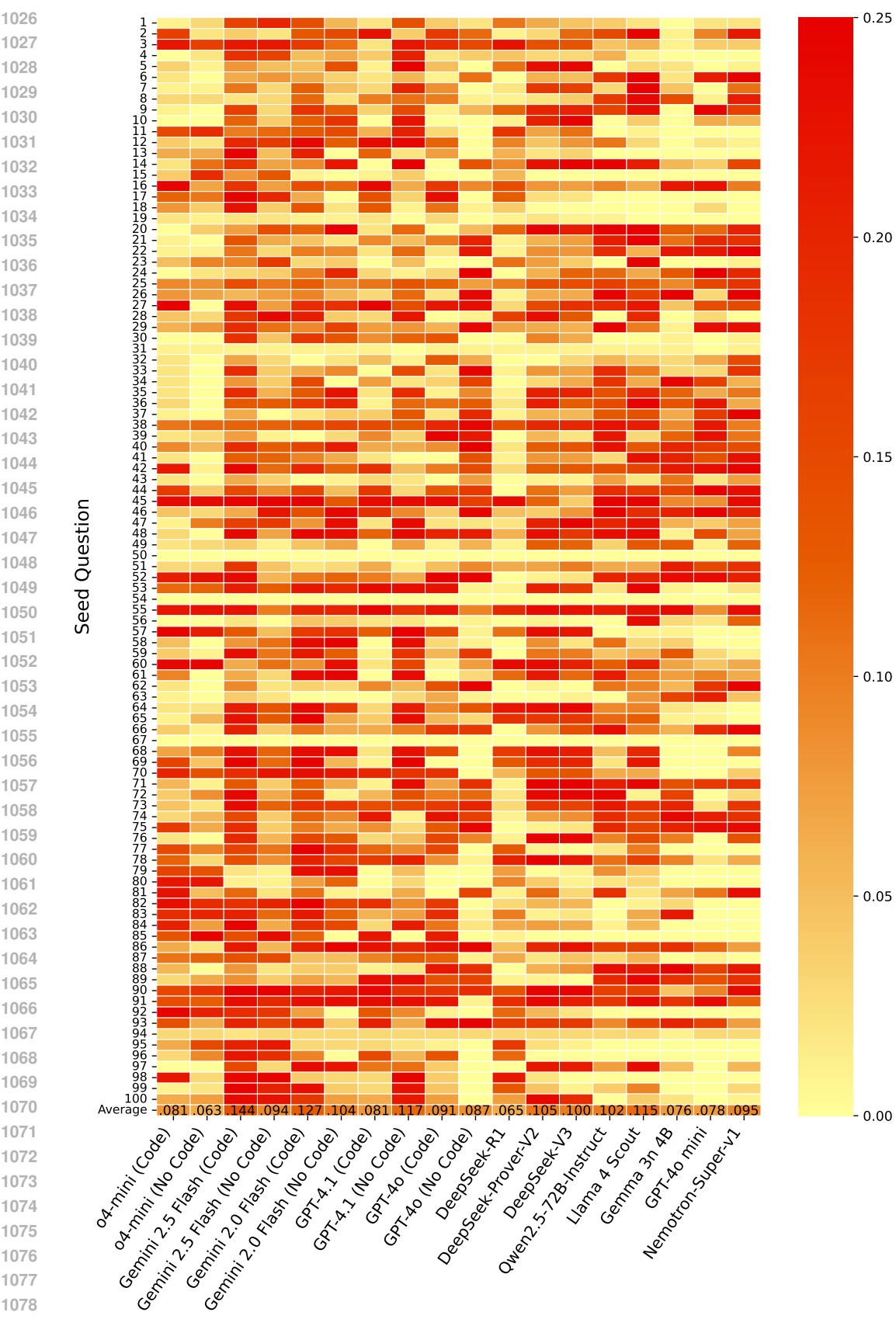

Figure 8: **Variance per model per seed question.** The bottom row shows the average variance of each model across all questions.

