# OpenReview forum: "ASyMOB: Algebraic Symbolic Mathematical Operations Benchmark"
_ICLR.cc/2026/Conference — Submitted to ICLR 2026_

### Official Review · Reviewer_P3Tt · 2025-10-30

**Soundness:** 3
**Presentation:** 1
**Contribution:** 2
**Rating:** 4
**Confidence:** 4

**Summary:**

This paper proposes a novel benchmark for LLM mathematical reasoning, namely ASyMOB. This benchmark perturbs existing problems using symbolic, numeric, and equivalence-preserving transformations to ensure robust evaluation. Several key insights are identified to guide the future development of mathematical LLM.

**Strengths:**

- The benchmark is reasonable, as the symbolic and numeric versions can fully evaluate the ability of LLMs to address mathematical reasoning.
- The provided examples are well-motivated, as identifying cases where both LLMs and symbolic systems do not perform well can help guide further research directions.

**Weaknesses:**

- The novelty of this paper requires further clarification. As noted at the end of this paper, GSM-Symbolic has conducted similar research and reached comparable conclusions. Therefore, it is important for the authors to clearly articulate the unique contribution and positioning of this work within the field, especially given the prior work, i.e., GSM-Symbolic. The authors should carefully clarify the difference between their benchmark and existing work. Additionally, some closely related studies are missing from the discussion, such as [1], which also employs neuro-symbolic methods for data generation. Including and discussing such relevant literature would further strengthen the paper.
- The conclusions and findings presented in this paper are trivial. For example, the integration of code tools has already been discussed in [2], and this is not a significant discovery or a new insight within the domain of LLM reasoning. In fact, numerous studies on agentic workflows [3] have previously explored approaches that enable LLMs to utilize a variety of tools.
- The presentation of this paper has significant room for improvement. For example, the paper ends with a lot of blank space. Some technical details are missing, such as the details of equivalence-preserving transformations, which are not clearly presented.

[1] Zenan Li, Zhi Zhou, Yuan Yao, Xian Zhang, Yu-Feng Li, Chun Cao, Fan Yang, Xiaoxing Ma. Neuro-Symbolic Data Generation for Math Reasoning. NeurIPS 2024.

[2] Luyu Gao, Aman Madaan, Shuyan Zhou, Uri Alon, Pengfei Liu, Yiming Yang, Jamie Callan, Graham Neubig. PAL: Program-aided Language Models. ICML 2023.

[3] Jiayi Zhang, Jinyu Xiang, Zhaoyang Yu, Fengwei Teng, Xionghui Chen, Jiaqi Chen, Mingchen Zhuge, Xin Cheng, Sirui Hong, Jinlin Wang, Bingnan Zheng, Bang Liu, Yuyu Luo, Chenglin Wu. AFlow: Automating Agentic Workflow Generation. ICLR 2025.

**Questions:**

Please refer to the weaknesses section. Additionally,
1. The list of equivalence-preserving transformations provided in the appendix appears to be too limited. It is unclear whether this small set of transformations is sufficient to comprehensively evaluate LLM capabilities, or whether it merely identifies a few isolated cases that might temporarily "hack" or exploit LLM behavior.
2. The evaluation costs associated with this benchmark should be further clarified. As a benchmark, the evaluation process should be both easy and affordable for researchers to carry out. It should also be clear whether the benchmark is sufficiently inexpensive to allow for comprehensive evaluation, or whether results obtained from evaluating only a subset of the benchmark are truly representative of overall performance.

---

> ### Author Response · Authors · 2025-11-25
>
> - *”The provided examples are well-motivated, as identifying cases where both LLMs and symbolic systems do not perform well can help guide further research directions.”*
>
> We thank the reviewer for the kind words.
>
> ---
>
>  - *“As noted at the end of this paper, GSM-Symbolic has conducted similar research and reached comparable conclusions. Therefore, it is important for the authors to clearly articulate the unique contribution and positioning of this work within the field, especially given the prior work, i.e., GSM-Symbolic.”*
>
> One of ASyMOB’s perturbation types (numeric) is conceptually related to a perturbation type used in GSM-Symbolic.
> Beyond this limited overlap, many recent works evaluate the reasoning capabilities of frontier models. This aspect is not the central point of novelty of ASyMOB.
>
> Our work differs substantially from GSM-Symbolic and from other works:
>
>  - **Scope of mathematical content**: ASyMOB evaluates symbolic mathematics capabilities - such as integration, differential equations, limits etc. - which are entirely absent from GSM-Symbolic. Our work is the first to evaluate university-level symbolic problems at scale.
> For comparison, GSM-Symbolic is derived from the older, simpler, and saturated GSM8K dataset, and its perturbations remain at a comparable grade-school level of difficulty.
>
>  - **Breadth of perturbation types**: In addition to numeric perturbations, ASyMOB introduces a range of new kinds of perturbations. In fact, we propose completely new categories for perturbation. Specifically, Symbolic and Equivalence perturbations fundamentally differ from the previous numeric perturbations and probe complementary dimensions of robustness (these have no counterpart in GSM-Symbolic).
>
>  - **Focus on mathematical reasoning rather than linguistic variation**: ASyMOB targets the mathematical reasoning core of each problem. Previous benchmarks examined the ability to interpret English text and translate it into formal mathematics. For example, GSM-Symbolic had most of its perturbations applied to the problem’s text. In fact, GSM-Symbolic’s strongest observed effect is linguistic rather than mathematical: “By adding seemingly relevant but ultimately irrelevant information to problems, we demonstrate substantial performance drops (up to 65%)...”
>
> We thank the reviewer for raising this point, and we now revise the introduction to clarify the novelty of our work relative to existing literature.
>
> ---
>
>  - *”Additionally, some closely related studies are missing from the discussion, such as [1], which also employs neuro-symbolic methods for data generation. Including and discussing such relevant literature would further strengthen the paper.”*
>
> We thank the reviewer for highlighting the work of Zenan Li, et al. NeurIPS 2024, which is now cited in the revised manuscript.
>
> Like GSM-Symbolic, this paper focuses on generating math word-problem variants - primarily drawing on GSM8K and MATH - in which symbolic patterns are encoded in SMT-LIB, perturbed within the symbolic space, and then translated back into natural language. In contrast, ASyMOB removes the verbal layer and introduces considerably higher-level symbolic mathematics. It also incorporates additional perturbation types beyond those considered in prior works.
>
> We now cite and discuss this work in the Introduction, also emphasizing the differences.
>
> ---
>
>  - *”...the integration of code tools has already been discussed in [2], and this is not a significant discovery or a new insight within the domain of LLM reasoning. In fact, numerous studies on agentic workflows [3] have previously explored approaches that enable LLMs to utilize a variety of tools.”*
>
> Indeed, the integration of code tools has already been explored in various works. We have cited key works in this domain and highlighted that contribution. We now also cite the additional papers suggested by the referee. The integration of code tools is not a claim of novelty in our work. We did not make such a claim. Following this comment by the referee, we now further emphasize that this is an aspect that has received major attention in the literature already.
>
> Our choice to measure the influence of code tools came to add interesting context to the main novel contribution: the introduction of a university-level symbolic math benchmark, with a range of new perturbations, and with a clear validation of success independent of language processing. We now clarify this fact in the revised manuscript and thank the referee for this clarification.
>
> ---
>
>  - *”...the paper ends with a lot of blank space…”*
>
> The blank space following the References section is deliberate, ensuring that the Appendix begins on a new page.
>
> With the new additions to the manuscript, the white space before the reference section is no longer there.
>
> ---

---

> > ### Author Response · Authors · 2025-11-25
> >
> > - *”Some technical details are missing, such as the details of equivalence-preserving transformations, which are not clearly presented”*
> >
> > We would appreciate a clarification regarding this comment. What specific technical details should be added?
> >
> > ---
> >
> >  - *”The list of equivalence-preserving transformations provided in the appendix appears to be too limited. It is unclear whether this small set of transformations is sufficient to comprehensively evaluate LLM capabilities, or whether it merely identifies a few isolated cases that might temporarily "hack" or exploit LLM behavior.”*
> >
> > The specific identities we selected were chosen without reference to model behavior or prior performance, and thus do not constitute “hacks” or targeted exploits.
> >
> > The identities in our list of Equivalence perturbations were selected specifically to be ones solvable by the evaluated models when presented in isolation. The idea was not to test the simplification abilities of LLMs here, but to evaluate them under an increased structural complexity of the full expression.
> > We agree that it is a worthy effort to study the simplification capabilities of LLMs (e.g., in comparison with Mathematica’s FullSimplify or SymPy’s Simplify). This study will require a comprehensive range of identities. Of course this test should be done separately from the ability to directly solve integrals and limits, as is the central point of our work here.
> >
> > This point will be clarified in Appendix A.1 and where we introduce the Equivalence perturbation.
> >
> > ---
> >
> >  - *”The evaluation costs associated with this benchmark should be further clarified. As a benchmark, the evaluation process should be both easy and affordable for researchers to carry out. It should also be clear whether the benchmark is sufficiently inexpensive to allow for comprehensive evaluation, or whether results obtained from evaluating only a subset of the benchmark are truly representative of overall performance.”*
> >
> > We thank the referee for this comment. The benchmark is sufficiently affordable for researchers to carry out. We add to Appendix B the following table, with the breakdown of evaluation costs per setting (prices vary mostly due to the vendor and the cost of tool use):
> >
> > | Model Name | Cost |
> > | :--- | :--- |
> > | **Closed-Weights Models** | |
> > | gemini-2.0-flash (no code) | 33$ |
> > | gemini-2.0-flash (code) | 32$ |
> > | gemini-2.5-flash (no code) | 687$ |
> > | gemini-2.5-flash (code) | 648$ |
> > | gpt-4.1 (no code) | 524$ |
> > | gpt-4.1 (code) | 1574$ |
> > | gpt-4o (no code) | 545$ |
> > | gpt-4o (code) | 1595$ |
> > | gpt-4o-mini | 16$ |
> > | o4-mini (no code) | 799$ |
> > | o4-mini (code) | 1849$ |
> > | | |
> > | **Open-Weights Models** | |
> > | gemma-3n-e4b-it | 22$ |
> > | meta-llama/Llama-4-Scout-17B-16E-Instruct | 273$ |
> > | nvidia/Llama-3_3-Nemotron-Super-49B-v1 | 250$ |
> > | Qwen/Qwen2.5-72B-Instruct | 261$ |
> > | DeepSeek-Prover-V2-671B | 244$ |
> > | DeepSeek-R1 | 927$ |
> > | DeepSeek-V3 | 244$ |
> >
> >
> > ASyMOB’s testing cost is kept in check by using a single LLM call per problem, unlike many prior works, which assess models using protocols that involve multiple calls per problem. For instance, Lewkowycz et al. (2022) evaluate Minerva on the MATH dataset (Hendrycks et al., 2021; 12.5K problems) using maj1@k majority voting with k values up to 256, resulting in a minimum of 3.2 million LLM calls.
> >
> > What we implemented is roughly 90X more cost-efficient due to our use of pass@1.
> > The reasoning is explained in Section 2.2 of the main text: “We exclusively use the pass@1 evaluation criterion, reflecting the practical requirement for reliability in real-world applications by engineers and researchers. The inherent LLM randomness is accounted for by evaluating success across the large number of questions within each category.”
> >
> > In addition, the “Numeric-All-N-S” perturbation category is designed primarily to evaluate stability and consistency. If these properties are less relevant to a given use case, evaluators may omit this category, reducing the total cost by 10,000 calls (about 28% of the full evaluation).
> >
> > This discussion is now also added to Appendix B.
> >
> > ---

---

### Official Review · Reviewer_eDb2 · 2025-11-01

**Soundness:** 2
**Presentation:** 2
**Contribution:** 2
**Rating:** 4
**Confidence:** 3

**Summary:**

This paper presents ASyMOB (Algebraic Symbolic Mathematical Operations Benchmark), a large-scale benchmark for evaluating large language models (LLMs) on symbolic mathematical reasoning. ASyMOB contains 35,368 verified problems generated from 100 seed questions through systematic symbolic, numeric, and equivalence-preserving perturbations, covering integration, limits, differential equations, series, and hypergeometric functions. These perturbations maintain mathematical equivalence while testing robustness to structural and numerical variation. The authors evaluate multiple open- and closed-weight LLMs (e.g., GPT-4o, Gemini 2.5, DeepSeek-R1) and find substantial performance degradation under minor perturbations (average accuracy drops from 74.6% to 46.8%).  However, frontier models show notably higher robustness, suggesting an emerging transition from memorization to true symbolic generalization. Additionally, ASyMOB reveals that LLMs sometimes outperform traditional CAS systems, and that hybrid LLM + CAS approaches can solve problems unsolvable by either alone.

**Strengths:**

- ASyMOB isolates symbolic mathematical reasoning from linguistic understanding, providing a clean test of algebraic manipulation skills.
- The symbolic, numeric, and equivalence perturbations enable fine-grained evaluation of robustness and generalization.
- Dual symbolic–numeric verification ensures reliability, and the findings reveal meaningful trends such as a "phase transition" toward genuine reasoning in frontier LLMs.

**Weaknesses:**

- The scope is somehow limited. The benchmark focuses narrowly on algebraic operations, omitting other mathematical reasoning domains, such as geometry or proofs.
- Some generated variants may be mathematically artificial and not representative of real-world symbolic problems.
- Several key conclusions, such as the role of code integration and hybrid tool use in improving LLM reasoning, have already been explored in prior work on tool-augmented or agentic LLMs, making the contributions more incremental than novel.

**Questions:**

1. How would the proposed perturbation framework generalize to other mathematical domains, such as geometry, proofs, or word problems that involve both symbolic and linguistic reasoning?
2. The paper interprets the robustness of frontier models as evidence of a "phase transition" from memorization to genuine symbolic reasoning. However, how do the authors rule out the possibility that these models simply memorize or interpolate over a much larger region of symbolic patterns, rather than performing true reasoning?

---

> ### Author Response · Authors · 2025-11-25
>
> - ASyMOB isolates symbolic mathematical reasoning from linguistic understanding, providing a clean test of algebraic manipulation skills.
>  - The symbolic, numeric, and equivalence perturbations enable fine-grained evaluation of robustness and generalization.
>  - Dual symbolic–numeric verification ensures reliability, and the findings reveal meaningful trends such as a "phase transition" toward genuine reasoning in frontier LLMs.
>
> We thank the reviewer for the kind words.
>
> ---
>
>  - *“The benchmark focuses narrowly on algebraic operations, omitting other mathematical reasoning domains, such as geometry or proofs.”*
>
> Early benchmarks in LLM evaluation sought to broadly assess problem-solving abilities across STEM domains. Over the past year, the community has recognized that such general-purpose benchmarks are overly broad, as they conflate fundamentally distinct cognitive skills. For instance, solving a geometry problem often relies heavily on visual perception and image-processing capabilities, whereas many mathematical or physics problems depend on natural language understanding. These differ substantially from pure symbolic manipulation skills, which come into play only after a problem has been correctly translated into a formal mathematical expression.
>
> This heterogeneity in required abilities has highlighted the need for more targeted evaluation protocols that isolate specific LLM capabilities.
> To address this gap, we built ASyMOB, providing an accurate measurement of LLM skills in symbolic math.
>
> The community is now shifting away from field-dependent (physics, chemistry, math, etc) and subfield-dependent benchmarks toward skill-specific evaluation frameworks. ASyMOB exemplifies this trend by focusing exclusively on symbolic-math reasoning.
>
> In this context, ASyMOB is most appropriately compared with benchmarks such as MiniF2F (Zheng et al., 2022) and MathConstruct (Balunović et al., 2025), which target formal theorem proving, or GeoQA (Chen et al., 2021), GeoEval (Zhang et al., 2024), and GeoSense (Gao et al., 2025), which isolate geometric reasoning abilities.
>
> ---
>
>  - *”Some generated variants may be mathematically artificial and not representative of real-world symbolic problems.”*
>
> While most ASyMOB perturbations do represent realistic scenarios, certain perturbations (such as very large numerical variants or the Equivalence-Hard category) are indeed “contrived”, in the sense of having a form that is more complex than provided in conventional exams and in other benchmarks. These questions are especially important for testing LLM abilities to generalize, as they surely do not appear in any training set.
>
> There is a second reason for adding such “over-complex” questions, and it is for testing the ability to **think in steps**. While questions provided in conventional exams would normally not be so complex, intermediate steps in long calculations often create such complex expressions that require symbolic simplification (especially after parameter changes, integration by parts, etc). The ability to simplify complex sub-expressions before attempting to directly solve the complete problem, is itself a **skill** worth testing.
>
> ---

---

> ### Author Response · Authors · 2025-11-25
>
> - *“Several key conclusions, such as the role of code integration and hybrid tool use in improving LLM reasoning, have already been explored in prior work on tool-augmented or agentic LLMs, making the contributions more incremental than novel.”*
>
> We are aware of the prior literature on code integration and hybrid tool use for mathematical reasoning, but our contribution is different in its novelty. Let us explain.
>
> We cited the major contributions in this area, including Novikov et al. 2025; Yue et al. 2024; A. Zhou et al. 2024; OpenAI 2025c; Liao et al. 2024; Gou et al. 2024; Imani et al. 2023; Romera-Paredes et al. 2023; and Dugan et al. 2024. This body of work, in fact, motivated our investigation of how code execution influences performance in symbolic-math problems.
>
> These are the primary papers that discussed the use of LLM-created code for mathematical problem solving - yet all of them focused on school-level math word problems. None of them focused on symbolic math, or measured performance on challenges like limits, differential equations or hypergeometric functions.
> Furthermore, the effect of question perturbations on LLM+code performance, and specifically the resilience of such systems to different types of perturbations, was never tested. All past works used fixed, often contaminated, benchmarks.
>
> Our findings on the effect of tool use can be distilled into three points:
>
> 1. Code execution generally increases robustness to problem perturbations - which could have been expected based on prior work, but never directly observed before ASyMOB.
> 2. For frontier models, however, tool use yields only marginal gains despite the benchmark not being saturated - a result that is both unexpected and new.
> 3. Certain hybrid tool-use strategies improve performance in ways that exceed the benefits of straightforward code execution by the LLM - an observation that is novel in the context of symbolic mathematics.
>
> Thus, even in the specific domain of LLM tool-use analysis, the work introduces contributions beyond what is available in prior research.
>
> ---
>
>  - *“How would the proposed perturbation framework generalize to other mathematical domains, such as geometry, proofs, or word problems that involve both symbolic and linguistic reasoning?”*
>
> This is an excellent question. The ASyMOB methodology is applicable across multiple domains. First on the conceptual level, the idea of a perturbation as an inherent part of a benchmark is becoming critical given the increasing issues of benchmark contamination. In fact, we are now developing a similar perturbation-based dataset for other domains, such as quantum-physics reasoning tasks.
>
> Regarding a more specific generalization of our framework to other domains, consider geometric problem solving. The three ASyMOB perturbation categories can be translated to geometry in a natural way:
>
>  - **Symbolic perturbations**: Geometric quantities such as lengths or angles can be replaced or scaled by symbolic variables. To ensure that the perturbed problems remain solvable and at the same conceptual level as the seed question, the substitutions should preserve structural relations. For example, if two angles are equal in the original problem, they must be replaced by the same symbol.
>
>  - **Numeric perturbations**: Analogously, geometric quantities can be replaced or scaled by new numeric values. To maintain geometric validity (for instance, the requirement that the interior angles of a triangle sum to $180^{\circ}$), fractional scaling is often more appropriate than arbitrary substitutions.
>
>  - **Equivalence perturbations**: As mentioned above, these are partly intended to introduce intermediate subproblems that must be resolved before the main solution can be reached, thereby testing multi-step reasoning. In geometry, this can be implemented by replacing a given quantity with an auxiliary construction that implicitly defines it. For example, a stated segment length could be replaced by a right triangle using that segment as the hypotenuse, while explicitly providing the lengths of the other two sides.
>
> An adaptation of ASyMOB to geometric tasks would therefore be conceptually straightforward, adheres closely to the original methodology, and would require minimal adjustments to the underlying code framework.
>
> ---

---

> > ### Author Response · Authors · 2025-11-25
> >
> > - *“The paper interprets the robustness of frontier models as evidence of a "phase transition" from memorization to genuine symbolic reasoning. However, how do the authors rule out the possibility that these models simply memorize or interpolate over a much larger region of symbolic patterns, rather than performing true reasoning?”*
> >
> > Although frontier models have almost certainly encountered the original seed questions during training, the randomized generation procedure used by ASyMOB makes it extremely unlikely that memorization or straightforward interpolation played any meaningful role in model performance.
> >
> > For instance, seed question #6 (appearing in Table 1) admits $10^9$ distinct Numeric-All-3-S variants, from which only 50 are sampled. The probability that any of these exact symbolic instances existed in pre-training corpora is effectively negligible. Even if we take an extreme case and assume that there are 100 symbolic problems with the exact structure as the seed question, and with 3-digit coefficients (as our Numeric-All-3 perturbations) in the training data, the probability of the LLM seeing even a single benchmark question is still only $\sim 5 \cdot 10^{-6}$ - ruling out direct memorization.
> >
> > Regarding *interpolation*, while it might be possible for the small numeric cases, the LLM would have to apply *extrapolation* - reasoning beyond its training distribution - to tackle larger numeric perturbations, most equivalence perturbations etc.
> > Such extrapolation is certainly a form of generalization by the LLM, thus providing evidence for reasoning.

---

### Official Review · Reviewer_ScbR · 2025-11-01

**Soundness:** 3
**Presentation:** 2
**Contribution:** 2
**Rating:** 4
**Confidence:** 3

**Summary:**

This work introduces ASyMOB, a 35,368 problem benchmark for symbolic mathematics (integration, limits, differential equations, series, etc.) built to test symbolic reasoning by systematically perturbing seed problems and then validating answers via equivalence-aware
symbolic and numeric checks. For every seed problem, they obtain variants of controlled difficulty through three methods: 1) symbolic perturbations, which perturb N symbols; numeric variants, which replace all or exactly one symbol with N-digit integers; and equivalence variants, which insert identities equal to 1 (trigonometric, hyperbolic, logarithmic, complex-exponential, and series). They further note that the dataset can be re-generated before assessing a new LLM, making their dataset more resilient against benchmark hacking or memorization. Finally, they find that 1) models' performance drops even under small perturbations, with frontier models being more robust, 2) incorporating code tools stabilizes performance, especially for weaker models, and 3) they observe instances where CAS fails but LLMs succeed and problems that can only be solved by combining both.

**Strengths:**

The work clearly explains how problems are built and expanded into symbolic, numeric, and equivalence variants, with worked examples for each.
* ASyMOB fills gaps in existing literature dataset, targeting symbolic manipulation (integration, limits, DEs, series, hypergeometrics) rather than text-to-math. It offers controlled difficulty via systematic perturbations and broad university-level problem coverage that previous benchmarks lack.
* Dataset instances are created with random transforms, and the dataset can be re-generated before evaluation. This design reduces leakage/memorization risk compared to static test sets.
* The work illustrates that models’ performance degrades sharply under small perturbations, while frontier models are more robust. Tool use helps weaker models, and hybrid LLM + CAS solves cases where either alone fails.

**Weaknesses:**

* The work documents qualitative examples where CAS fails but LLMs succeed, and a case solvable only by an LLM + CAS hybrid (Figure 6). Further, it argues that symbolics hurt CAS more than LLMs. What’s missing is a dataset-level percentage/table partitioning successes into LLM-only, CAS-only, and hybrid categories across perturbations. Adding this would substantively strengthen the claim.
* Some of the perturbations appear to be somewhat contrived. This may not necessarily be a bad thing, but it could mean that the benchmark is testing perturbations that would not reasonably appear in the real world. This is related to my first question below.

**Questions:**

* Does performance on ASyMOB correlate with performance on other real-world tasks? E.g., performance on math contests that occurred after model releases.
* Would training/fine-tuning on ASyMOB perturbations lead to stronger models?

---

> ### Author Response · Authors · 2025-11-25
>
> - *The work clearly explains how problems are built and expanded into symbolic, numeric, and equivalence variants, with worked examples for each.*
>  - *ASyMOB fills gaps in existing literature dataset, targeting symbolic manipulation (integration, limits, DEs, series, hypergeometrics) rather than text-to-math. It offers controlled difficulty via systematic perturbations and broad university-level problem coverage that previous benchmarks lack.*
>  - *Dataset instances are created with random transforms, and the dataset can be re-generated before evaluation. This design reduces leakage/memorization risk compared to static test sets.*
>  - *The work illustrates that models’ performance degrades sharply under small perturbations, while frontier models are more robust. Tool use helps weaker models, and hybrid LLM + CAS solves cases where either alone fails.*
>
> We thank the reviewer for the kind words.
>
> ---
>
>  - *“The work documents qualitative examples where CAS fails but LLMs succeed, and a case solvable only by an LLM + CAS hybrid (Figure 6). Further, it argues that symbolics hurt CAS more than LLMs. What’s missing is a dataset-level percentage/table partitioning successes into LLM-only, CAS-only, and hybrid categories across perturbations. Adding this would substantively strengthen the claim.”*
>
> This is a great point. Following this suggestion, we tested the performance of SymPy, as that is the tool which code-integrated LLMs are using.
> We compare this CAS-only performance to the results we had on the LLM-only and hybrid categories.
>
> CAS-only performance analysis requires distinguishing between the failure modes of CASs, which differ from those of LLMs. When a CAS produces a result, it is - barring extremely rare exceptions - correct. The dominant failure mode is therefore not incorrect output but an inability to complete the symbolic task: the system may return an expression that remains unresolved (e.g. still containing an integral or infinite series) or fail to return within the allotted runtime.
>
> ASyMOB was designed to expose limitations in stepwise symbolic reasoning, with the Equivalence perturbation category being particularly demanding. These hierarchical symbolic transformations are especially problematic for SymPy. For instance, among the 280 Equivalence variants derived from seed question #16, 264 ended in timeouts (94%). Under a relatively modest timeout of five minutes, this rate of failure implies that evaluating only the Equivalence portion of the benchmark requires roughly 1,500 hours - approximately 62.5 days. LLMs are much less susceptible to such runtime issues.
>
> In contrast to CAS performance on the Equivalence variants, its performance is fairly robust under Numeric perturbations.
>
> Using a representative subset of the benchmark, we observed the following accuracy levels:
>
> | Category | Success Percentage | Ranking Compared to Other Tested Settings |
> | :--- | :--- | :--- |
> | **Symbolic** | 56.7% | 13/18 |
> | **Numeric** | 65.2% | 8/18 |
> | **Equivalence** | 21.9% | 14/18 |
>
> Table 2 in the paper will now include a line detailing the relative performance of SymPy.
>
> ---
>
>  - *“Some of the perturbations appear to be somewhat contrived. This may not necessarily be a bad thing, but it could mean that the benchmark is testing perturbations that would not reasonably appear in the real world. This is related to my first question below.”*
>
> While most ASyMOB perturbations do represent realistic scenarios, certain perturbations (such as very large numerical variants or the Equivalence-Hard category) are indeed “contrived”, in the sense of having a form that is more complex than provided in conventional exams and in other benchmarks. These questions are especially important for testing LLM abilities to generalize, as they surely do not appear in any training set.
>
> There is a second reason for adding such “over-complex” questions, and it is for testing the ability to **think in steps**. While questions provided in conventional exams would normally not be so complex, intermediate steps in long calculations often create such complex expressions that require symbolic simplification (especially after parameter changes, integration by parts, etc). The ability to simplify complex sub-expressions before attempting to directly solve the complete problem, is itself a **skill** worth testing.
>
> ---

---

> ### Author Response · Authors · 2025-11-25
>
> - *“Does performance on ASyMOB correlate with performance on other real-world tasks? E.g., performance on math contests that occurred after model releases.”*
>
> This is an interesting question.
> We see a general correlation to benchmarks such as AIME 2025 (taken from the Kaggle leaderboard):
>
> | Model | AIME 2025 Score |
> |------------------------|-----------------|
> | **o4-Mini** | 82.5% |
> | **GPT-4.1** | 36.7% |
> | **GPT-4o-Mini** | 8.3% |
> | **GPT-4o** | 5.8% |
> | **Gemini-2.5 Flash** | 75.8% |
> | **Gemini-2.0 Flash** | 29.2% |
> | **DeepSeek-V3** | 25.0% |
> | **DeepSeek-R1** | 84.2% |
>
> A very relevant comparison is to RIMO, which just came out 2 months ago [Chen et al., arXiv:2509.07711], evaluating LLM performance on mathematical contest problems. RIMO reported results that correlate with those observed in ASyMOB for the three models common to both benchmarks:
>
> | Model Name | Performance |
> | :--- | :--- |
> | **DeepSeek-R1-671B** | 63% |
> | **Gemini-2.5-flash** | 58.8% |
> | **GPT-4o-2024-08-06** | 33.4% |
>
> Surprisingly, on PutnamBench - a benchmark targeting formal mathematical reasoning - DeepSeek-Prover-V2-671B achieved 47/668 (pass@1024), while DeepSeek-V3-0324 and Gemini-2.5-pro-exp-0325 obtained 0/668 (pass@1) and 3/668 (pass@1), respectively - not showing correlation to ASyMOB results.
>
> These results indicate that strong formal-proof performance does not necessarily transfer to the broader domain of symbolic mathematical manipulation, where general reasoning ability appears more predictive - particularly for Equivalence perturbations that require multi-step structural analysis.
>
> These comparisons are now discussed and cited in Section 3.
>
>
> ---
>
> - *“Would training/fine-tuning on ASyMOB perturbations lead to stronger models?”*
>
> Yes. In Section 4, where we briefly discuss this idea, we now also point to a previous paper [Li et al. from NeurIPS 2024] showing that adding perturbation-based synthetic data to the training improves the overall LLM performance in math:
>
> “Looking forward, LLMs should be intentionally trained to generalize, both via tool use and through systematic perturbations on the training set. Previous works showed that such synthetic  data improves overall performance (Li et al. 2024). Fine-grained perturbations provide a systematic method for generating high-quality synthetic data, offering a valuable resource for fine-tuning future reasoning models.”
>
> Specifically, one could fine-tune an LLM by identifying and isolating performance discrepancies: cases where the model succeeded on the seed question but failed on a perturbed version. Such high-resolution performance information could then be utilized for fine tuning. In addition, regeneration of variations during each training epoch would serve as a mechanism to mitigate overfitting and promote generalization.

---

### Meta-Review · Area_Chair_kYst · 2025-12-27

**Summary:**

This paper targets to build a benchmark for symbolic mathematics. After reviewing all comments and the rebuttal,  this paper does not reach the room for acceptance due to several key concerns which limit its suitability.

Most importantly, reviewers questioned the *novelty and contribution* of the work. The perturbation-based evaluation framework and findings on tool use and hybrid LLM–CAS reasoning substantially overlap with prior benchmarks and existing literature, making the overall contribution appear incremental. Additionally, several core claims, most notably the interpretation of improved robustness as a “phase transition” from memorization to genuine reasoning, were viewed as *insufficiently substantiated*, as alternative explanations such as large-scale interpolation could not be convincingly ruled out.

Reviewers also raised concerns about the *representativeness and practical relevance* of the benchmark, noting that some perturbations appear mathematically artificial and may not reflect realistic symbolic reasoning tasks. Finally, issues related to *presentation clarity and evaluation cost* further detracted from the paper’s maturity.

Overall, these concerns point to a core weakness of the paper: the proposed benchmark does not sufficiently demonstrate a clear or distinctive contribution relative to existing benchmarks in the literature. Taken together, these issues support the decision to reject.

**Reviewer Concerns:**

After reading the rebuttal, the main concerns related to conceptual or contribution discussions may not be addressed.

**Reviewer Scores:**

After reading the rebuttal, I do not expect the reviewers to substantially change their scores. The rebuttal primarily responds to the reviewers’ concerns through discussion and clarification, but it does not fundamentally address the core issue regarding the benchmark’s unique contribution relative to existing work. As such, these concerns cannot be resolved through argumentation alone.

---

### Decision · Program_Chairs · 2026-01-26

Reject